# The impact of light heterogeneity in controlled environment agriculture on biomass of microgreens

**Original Research Article**

vertical farms; microgreens; optimisation; brassica; light sensitivity.

**Corresponding author:**
Daphne Ezer;
Email: daphne.ezer@york.ac.uk

**Associate Editor:**
Prof. Boon Leong Lim

Will Claydon[1], Phoebe Sutton[3], Ethan J. Redmond[1], Gina Y.W. Vong[1],

Alana Kluczkovski[1,2], Alice Thomas[1,2], Katherine Denby[1,2] and Daphne Ezer[1]

[1]Department of Biology, University of York, York, UK; [2]Centre for Novel Agricultural Products (CNAP), Department of Biology, University of York, York, UK; [3]Vertically Urban, Typhoon House, Leeds, UK

## Abstract

Yield is impacted by the environmental conditions that plants are exposed to. Controlled environmental agriculture provides growers with an opportunity to fine-tune environmental conditions for optimising yield and crop quality. However, space and time constraints will limit the number of experimental conditions that can be tested, which will, in turn, limit the resolution to which environmental conditions can be optimised. Here we present an innovative experimental approach that utilises the existing heterogeneity in light quantity and quality across a vertical farm to evaluate hundreds of environmental conditions concurrently. Using an observational study design, we identify features in light quality that are most predictive of biomass in different kinds of microgreens (kale, radish and sunflower) that may inform future iterations of lighting technology development for vertical farms.

## 1. Introduction

Plants are sensitive to minor changes in environmental conditions, such as light, temperature and humidity. For instance, Arabidopsis is sensitive to as little as a 2°C change in temperature (Balasubramanian et al., 2006). Vertical farms are indoor farms in which multiple layers of planting beds are stacked on top of each other, exploiting the vertical space to enable more crops to be grown per square meter. Vertical farming is considered a kind of controlled environmental agriculture (CEA) practice, because it is possible for growers to customise the growing conditions of crops to maximise yield for each species by controlling aspects of the plant's environment such as their temperature and light exposure (Farhangi et al., 2023), see Figure 1a. Minor environmental (Gavhane et al., 2023; Ke et al., 2021; Kong & Nemali, 2023) differences may also impact agriculturally important qualities other than yield, such as aesthetic qualities, nutrient concentration and taste compounds (Gavhane et al., 2023; Ke et al., 2021; Kong & Nemali, 2023). Farmers need to balance these requirements when designing their optimised vertical farm conditions.

To optimise vertical farm conditions, researchers will usually employ a complete randomised experimental design in which they measure crop traits in several different conditions, a strategy that has been successfully deployed in many studies to optimise light, temperature and humidity separately (Carotti et al., 2020; Most et al., 2019; Wang et al., 2016). However, these studies do not consider how changing one growth condition may impact another. Other researchers have utilised multifactorial experimental designs, especially full factorial designs, to combinatorically assess the impact of several environmental variables concurrently (An et al., 2021; Ciriello et al., 2023; Kamenchuk et al., 2023). For instance, a 4 x 4 x 3 full factorial design was used to find optimal combinations of root temperature, air temperature and light intensity for lettuce growth (Carotti et al., 2020). In contrast, other groups wished to optimise across six environmental parameters at three levels, which would have required 729 experimental treatments using a full factorial design, but they were able to select 27 combinations of conditions to test by using the Taguchi Method to generate an Orthogonal

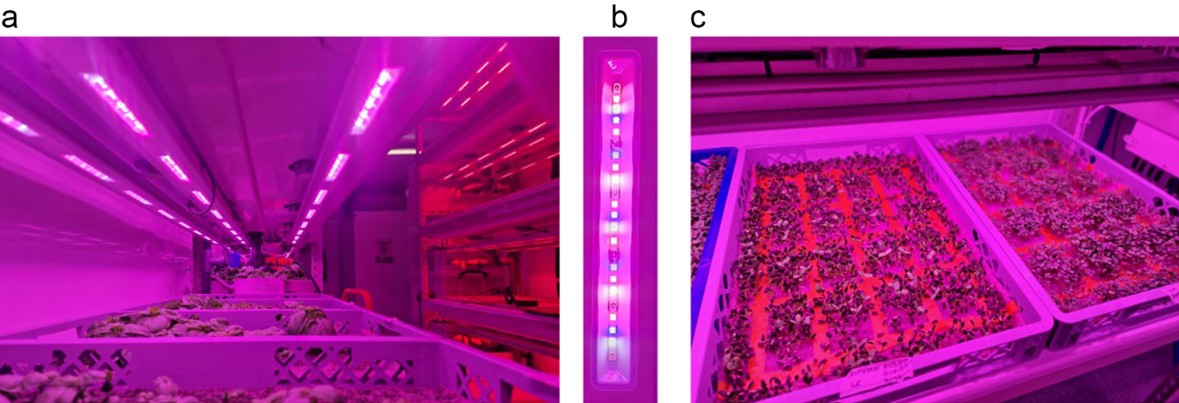

**Figure 1.** Photographs of the farm. (a) Arrangement of lights above each bay. (b) Close up of the LED lights. (c) Subdivisions of trays for experiment.

Array of treatment combinations (Farhangi et al., 2023). The Taguchi Method is a strategy of selecting a smaller number of combinations of conditions to experimentally test: instead of testing all possible combinations of all the parameters, the Taguchi Method ensures that all possible combinations of every pair of parameters are tested at least once. However, even the Taguchi method requires a relatively large number of treatment conditions, which would be difficult to implement in a small vertical farm. Moreover, any randomised experimental design will assume that treatments are homogenous within each treatment block, when in fact light, temperature and humidity may vary within the physical space due to the layout of lights (Figure 1) and the air flow, among other factors, microclimatic conditions that have been modelled in digital twins of vertical farms (Agati et al., 2024; González et al., 2022).

Although the physiological impact of heterogeneous microclimates in vertical farms has not been fully investigated, it has been widely established that in traditional field-based agriculture, conditions are not uniform within a field, resulting in heterogeneity in crops. Soil topography can vary across and within fields, which has been shown to impact how soil can accumulate water and by proxy impact yield (Maestrini & Basso, 2018). It has also been shown that increasing the distance from the edge of a field that a crop is grown in can increase yield. Moreover, crop yields can also be impacted by the landscape that surrounds a field and the amount of shading that a crop receives (Fincham et al., 2023). Within the field variation of the environment is detrimental to both food security and to profits made by farmers. This is because within-field variation impacts how a crop is shaped, its size, colour and yield, all factors that can lead to crops failing quality checks and being disposed of (Ishangulyyev et al., 2019). Vertical farming removes sources of heterogeneity such as soil topography and landscape. However, there are still sources of heterogeneity that occur within vertical farms as shown here.

In this paper, we have taken advantage of the heterogeneity of light intensity and quality that exists in a vertical farm to suggest an innovative observational experimental design for condition optimisation. We measured the position-specific light intensity and quality across 256 different positions in a vertical farm and modelled how these factors impact biomass of different kinds of microgreens. We suggest that exploiting existing heterogeneity in vertical farms to perform observational studies will enable us to optimise treatments using less space-intensive experimental set-ups than in traditional environmental optimization approaches,

allowing researchers to test hundreds of different combinations of conditions in a single experiment.

## 2. Materials and methods

### 2.1. Measuring light heterogeneity in a vertical farm

This work was performed in the Grow It York vertical farm (Doherty et al., 2022) which uses LettUs Grow aeroponic technology (Chittibomma et al., 2023) in central York. Plants were grown in 0.25 m$^2$ trays, with four trays per bed, with four beds stacked vertically, totalling 16 trays. Above each bed was a set of three LED lighting fixtures, Horti-blade BRWFR-4 spectrum, from Vertically Urban. Due to space constraints during retrofitting, the LED fixtures were positioned unequally above the bed with two fixtures towards the front and one towards the back.

All seed varieties and their sources are found in Table 1. To establish the extent of light heterogeneity within the vertical farm, we divided each tray into 16 unique sections each with an area of 83.7 cm$^2$. Microgreens were also grown in a ring which measured 9cm at the left and right sides of the mat and 11.5 cm at the top and bottom. These rings surrounded the 16 squares in the centre of the mat. The plants in these rings were not measured as part of this experiment to negate edge effects in our samples (see Supplementary Figure S1). Each position was given a unique *x* and *y* coordinate and in these same positions, plants would be grown and have their biomasses recorded. Across all beds, this resulted in 256 unique positions. Both photosynthetically active radiation (PAR) and spectral irradiance were measured once within a month of the experiment taking place in these positions at the level of the tray in the absence of plants. PAR was measured using a PAR Special from Skye Industries. Spectral irradiance was measured using an Ocean Fire Spectrometer from Ocean View.

### 2.2. Finding average blue, red and far-red light measurements

Plants were grown under a spectrum consisting of 22%**B** 14%**G** 64%**R** 7%**FR**, characterised by blue (400–500 nm), green (500–600 nm), red (600–700 nm) and far-red wavelengths (700–780 nm). For our experimental purposes, we measured narrower waveband ranges: 445–456 nm for blue, 653–668 nm for red and 725–735 nm for far red (Banerjee et al., 2007; Butler et al., 1964). These experimental wavebands correspond to plant photobiological sensitivity, and the LED emission peaks of the vertically urban lighting

**Table 1.** Description of microgreen varieties and harvest times

| Species | Common name | Source | Time to harvest after germination (days) |
|---|---|---|---|
| *Brassica oleracea* | Kale (Dwarf green curled) | Tozer Seeds | 7 |
| *Brassica oleracea* | Kale (Dwarf Blue) | CN Seeds | 7 |
| *Helianthus annuus* | Sunflower (Black Seeded) | CN Seeds | 7 |
| *Raphanus sativus* | Radish (Sangrin) | CN Seeds | 5 |

system (see Supplementary Figure S2). The average intensities were calculated for each experimental waveband at each of the 256 positions.

## 2.3. Observational experiment

All plants were grown under 12 hours of light per day which delivered 54.75–62.32 $\mu W/cm^2/nm$ of red light, 37.99–79.36 $\mu W/cm^2/nm$ of blue light and 5.43–14.43 $\mu W/cm^2/nm$ of far-red light. This corresponds to a PAR reading in the range of 263–568 $\mu mol\ m^{-2}\ s^{-1}$ for 12 hours a day. These light values are expressed as ranges due to the light heterogeneity in the facility (see Supplementary Table S1). PAR readings are in different units of measurement as spectral measurements, because they are values that are calculated based on the absorption spectra of plants and estimate the quantity of photons that are estimated to be absorbed by the plants per surface area each second. It is challenging to convert to $\mu W/cm^2/nm$, because conversion requires knowing the distribution of the wavelength of lights. On the other hand, vertical farming light engineers prefer the unit $\mu W/cm^2/nm$, because it directly relates to energy consumption. For this reason, we chose to maintain different sets of units for PAR and light quality measurements throughout.

Microgreens were grown on Growfelt wool carpet matting. An aeroponic system was used where the pH was 5.9 and the electrical conductivity was 1.7. The nutrient solutions used were Hydromax Grow A and B. The aeroponic system was activated for 1 minute every 5 minutes. Each microgreen variety, as outlined in Table 1, was grown in one tray per bed in the arrangement shown in Supplementary Figure S3. Please note that there are two varieties of kale, distinguished herein by their providers, Kale (CN) provided by CN Seeds and Kale (Tozer) provided by Tozer Seeds. This equalled four trays per variety giving 64 unique measurement positions per microgreen variety. All samples were grown for the period specified in Table 1 which varied by species. Each unique position was harvested individually and biomass was recorded.

## 2.4. Generating mixed effects models

Our aim was to predict the biomass of microgreens, as this is the yield metric. We used a mixed effects model to predict biomass per position (g/83.7 $cm^2$), using the lme4 package in R (T. Wang et al., 2022). We assumed that bed and position within a tray would both be confounding factors influencing biomass. We also assumed that the impact of bed and position within a tray would depend on the species/variety of microgreens. Our baseline model was as follows, where m is the microgreen species/variety (see Table 1), *b* is the bed and *c* is the position within the tray (i.e. centre, edge or corner, see Supplementary Figure S1).
Equation 1:

$$\text{Biomass}_m \sim \beta_0 + \beta_m + \beta_{b,m} + \beta_{c,m} + e_{b,c,m}$$

Next, we added light intensity (PAR, $\mu mol\ m^{-2}\ s^{-1}$) to the baseline model, allowing for light intensity to have a different effect on each microgreen variety.
Equation 2:

$$\text{Biomass}_m \sim \beta_0 + \beta_m + \beta_1 \text{PAR} + \beta_{2,m} \text{PAR} + \beta_{b,m} + \beta_{c,m} + e_{b,c,m}$$

Finally, we added light quality ($\mu W/cm^2/nm.$) to the baseline model, again assuming that each microgreen variety may have different sensitivity to light quality, where B, R and FR represent the mean light intensity in the blue, red and far red light quality bands ($\mu W/cm^2/nm$), as specified in the previous section.
Equation 3:

$$\text{Biomass}_m \sim \beta_0 + \beta_m + \beta_1 B + \beta_2 R + \beta_3 FR$$
$$+ \beta_{4,m} B + \beta_{5,m} R + \beta_{6,m} FR + \beta_{b,m} + \beta_{c,m} + e_{b,c,m}$$

As many plant light sensors detect ratios of light quality, we also developed a model that incorporates light quality ratios, where R:B, R:FR and FR:B represent the log10 ratios red (R), blue (B) and far red (FR).
Equation 4:

$$\text{Biomass}_m \sim \beta_0 + \beta_m + \beta_1 R:B + \beta_2 R:FR + \beta_3 FR:B$$
$$+ \beta_{4,m} R:B + \beta_{5,m} R:FR + \beta_{6,m} FR:B + \beta_{b,m} + \beta_{c,m} + e_{b,c,m}$$

To compare the performance of these models we used an approach published by (Zuur et al., 2009). The models with random effects were compared to the baseline model through separate ANOVAs, with all *p*-values compared to a 0.05 threshold after a Holm's correction for multiple hypothesis testing. The Bayesian Information Criterion and Akaike Information Criterion were also assessed.

## 2.5. Regularising general linear modelling

Equations 2–4 contain a large number of parameters compared to the number of observations, which introduces a risk of overfitting. We sought to use a well-established procedure to select a smaller set of parameters to include in our model. Specifically, a lasso regularisation procedure was employed to select a smaller subset of variables and prevent overfitting, using the glmnet package in R (Engebretsen & Bohlin, 2019). Data were standardised prior to fitting as a *z*-score (each variable was subtracted by its mean in the training data and divided by the standard deviation of the training data), but the coefficients reported here are adjusted to be in the original scale. The initial model specification took the form:
Equation 5:

$$\text{Biomass}_m \sim \beta_0 + \beta_m + \beta_1 \text{PAR} + \beta_{2,m} \text{PAR}$$
$$+ \beta_1 B + \beta_2 R + \beta_3 FR$$
$$+ \beta_{4,m} B + \beta_{5,m} R + \beta_{6,m} FR$$
$$+ \beta_7 R:B + \beta_8 R:FR + \beta_9 FR:B$$
$$+ \beta_{10,m} R:B + \beta_{11,m} R:FR + \beta_{12,m} FR:B$$
$$+ \beta_{b,m} + \beta_{c,m} + e_{b,c,m}$$

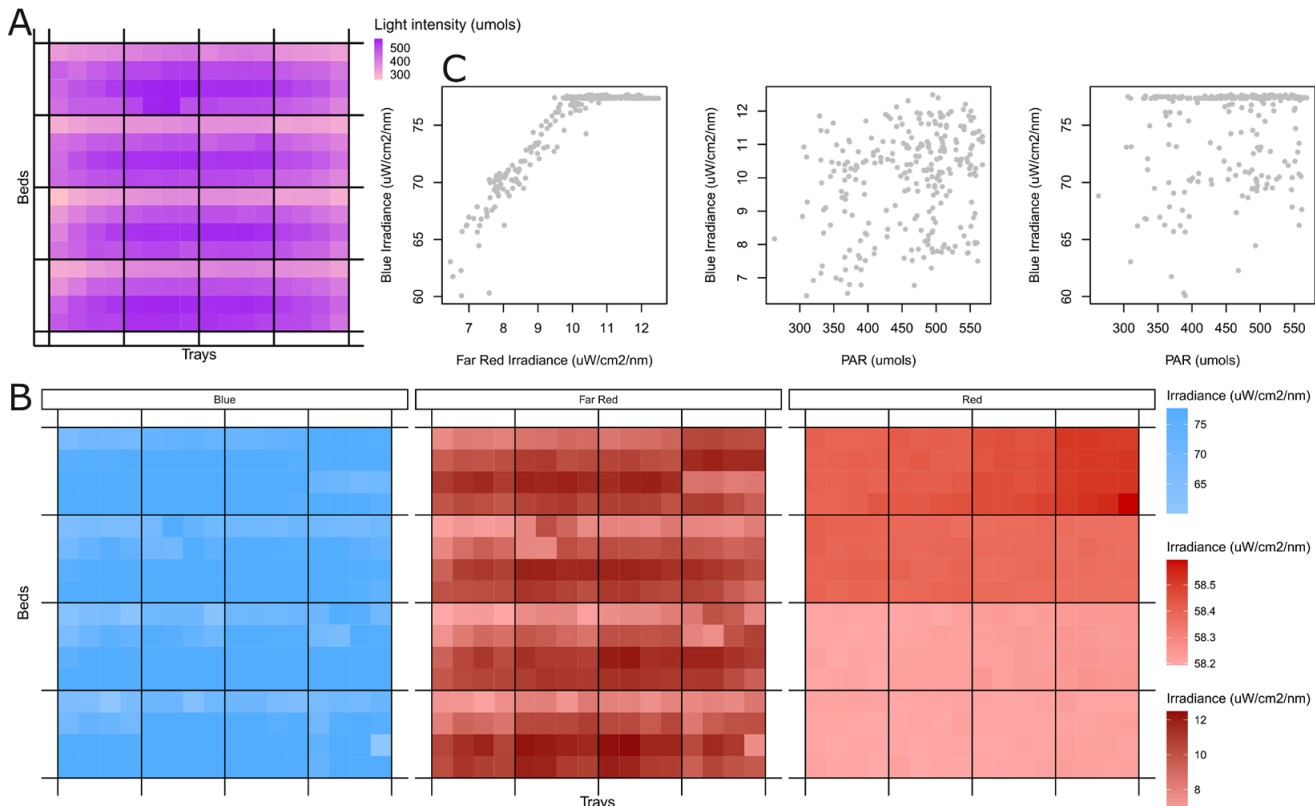

**Figure 2.** Heterogeneity of light quantity and quality within the vertical farm: (a) PAR readings and (b) average intensity within the following wavelength bands blue: 445–456, red: 653–668 and far-red: 725–735 are shown across the farm. The beds are vertically stacked and each contains four trays with 16 positions per tray. Each square represents an area of size 83.7 cm$^2$. Note that the plants grown along the edge of each tray are not included in this study and the light was not measured in these positions. (c) The relationship between blue and far-red irradiance and PAR readings, where each point represents a position/square in the images in (a) and (b).

This model was evaluated using a two-level leave-one-out cross-validation approach. For each of the 256 observations, a separate glmnet model was fit after leaving out one observation (i.e. 255 observations used to train the model). To select the lambda parameter of this model that would minimise the mean squared error, a further leave-one-out cross-validation strategy was deployed (using 254 observations for training the models for selecting the lambda parameter, by leaving out one of the 255 observations in the training set).

The output of this procedure was a set of 256 different models, each trained on 255 observations. For each model, we then predicted the biomass of the observation that was not used in training the model. This model-fitting approach allows us to evaluate model performance in a way that is not affected by overfitting by testing our models on observations that were not used to train the model. Moreover, this approach provides us with a collection of 256 fitted values per coefficient, which enables us to evaluate how reliant our coefficient predictions are on the inclusion of any individual data point.

## 3. Results

### 3.1. Characterising the heterogeneity of lights in a vertical farm

Our first aim was to quantify the level of light heterogeneity at plant level within the vertical farm. To investigate the extent of this heterogeneity we measured both PAR (Figure 2a) and spectral irradiance (Figure 2b) in 256 unique growing positions.

The light intensity (PAR reading) was highest towards the front and centre of each bed (Figure 2a). A similar pattern of light intensity was observed for blue and far red wavebands(Figure 2b), except for the rightmost trays in each bed. For red light, there was greater variation between beds than within beds, with the top two beds having higher levels than the bottom two (Figure 2b). Far-red and blue light intensities were highly correlated at low light settings, but blue light levels saturated at higher light intensities (Figure 2c). Although there was a significant correlation between light intensity and blue and far red light quality, (Pearson correlation test, $p<0.001$ for both), the Pearson correlation coefficient was low ($R = 0.22$ and $R = 0.21$, respectively). These results demonstrate the variation in light quantity and light quality in the vertical farm.

### 3.2. Light quality is a predictor of biomass

In order to determine whether light quantity, light quality or ratios of light quality were predictors of biomass (the yield metric for microgreens), mixed effect models were constructed and compared to a baseline model that did not include any light-related parameters. Statistics for these comparisons are available in Table 2. The best-performing model was the light quality-based model (Equation 3), performing significantly better than the baseline model, even after Holm's correction for multiple hypothesis testing ($p < 0.05$). It also minimised the Akaike Information Criterion (AIC); however, this model performed poorly under the Bayesian Information Criterion (BIC) as BIC places a greater penalty on the number of parameters in the model. These results suggest that light

**Table 2.** Summary of mixed effect model outcomes

| Model | AIC | BIC | *p* | *p* Adjust |
|---|---|---|---|---|
| Baseline | 1531.2 | 1556.0 | NA | NA |
| Intensity | 1529.7 | 1568.7 | 0.04987 | 0.09974 |
| Quality | 1526.0 | 1593.4 | 0.003724 | 0.01117 |
| Ratios | 1535.1 | 1588.2 | 0.1447 | 0.1447 |

quality can be used to predict biomass and that this is a better predictor of biomass than PAR readings alone.

This analysis was unable to determine whether combining information about light intensity and light quality ratios could further improve the model. In the next section, we will try to minimise the complexity of the model but also include combined information about light quality and quantity.

### 3.3. Variable selection highlights which aspects of light intensity and quality are most predictive of biomass

In order to determine which combination of explanatory variables was most predictive of biomass, we deployed a variable selection procedure, as described in the methods (via lasso regularisation). Our reduced models were able to accurately predict the biomass of samples that were not used in training the model (see Figure 2a), with Pearson's $R = 0.864$ and $P < 0.05$ ($P = 7.6e{-}78$). Our predictions were also significantly correlated with the true biomass ($p < 0.05$) for each of the individual varieties after Holm's correction, with Pearson's $R$ of 0.247, 0.487, 0.597 and 0.595 for kale (CN), kale (Tozer), radish and sunflower, respectively. Notably, the variety for which our model performed the worst was kale (CN), which was the lowest biomass variety and also the one with the least variance in biomass.

In our further analysis of model coefficients, we decided to include all explanatory variables that were selected in at least 80% of the models (see Supplementary Figure S4 and Supplementary Table S2). Some light-related variables were equally associated with biomass across all microgreen varieties: light intensity (via PAR reading) and red and blue light quality. Of these, red light quality was strongly negatively associated with biomass in all models, while blue light was positively associated (Figure 3b). Overall light intensity was slightly negatively correlated with biomass in all models (Figure 3b).

Although on their own, light quality ratios did not improve our ability to predict biomass compared to the baseline model, we found several light quality ratio variables that were selected in the combined model. However, these were all specific to individual varieties of microgreens. For instance, separate red:far-red ratio coefficients were selected for each microgreen, with negative associations for both kale varieties and positive associations for radish and sunflower (Figure 3c). A full table of predicted coefficients for all models is available in Supplementary Table S2.

### 3.4. Variable prioritisation suggests R:FR ratio is a target area in kale and radish production

To further prioritise aspects of light quality impacting yield heterogeneity, we decided to identify light features which were found to be associated with biomass independently in the top two beds and the bottom two beds. The motivation for this is that the top two beds and bottom two beds were found to contain large differences

in red light quality (Figure 2b), and we wanted to find variables that influenced biomass under these different red light conditions. To do this, we repeated our two-level lasso regularisation, using only the top two beds and the bottom two beds. The total number of variables selected was reduced, likely because we halved the number of observations used to train the model. However, the red-to-far red light ratio was selected as a key variable for radish and kale (T) in the top two beds, the bottom two beds and in the combined model, with a higher R:FR ratio associated with higher yields in radish and lower yields in kale (T) (Figure 3d). These results confirmed that the R:FR ratio has an opposite direction of association with radish and kale (T) and that this pattern is consistent across the two different red light treatments that these varieties were exposed to. This suggests that the R:FR ratio is an appealing target for further spectral refinement, but that this needs to be performed in a variety-by-variety manner.

## 4. Discussion

While there are some large vertical farming operations (Stein, 2021), many vertical farms (especially R&D vertical farms) are relatively compact, being situated in shipping containers (Schmidt Rivera et al., 2023), in distribution centres (Al-Kodmany, 2018) or even in retail centres (Martin et al., 2023). For this reason, it may be impractical for these farms to iteratively perform light optimisation experiments, especially factorial designs that investigate the interactions between variables. Moreover, the knowledge obtained from optimising one vertical farm may not translate into optimal conditions in a different setting, as the light-dependency on yield may be dependent on the specific vegetable variety being grown (Cammarisano & Körner, 2022) and other extrinsic conditions, like the temperature, humidity and growth medium (Carotti et al., 2020). For instance, it has been found that red light can increase or decrease yield, depending on the wider growth context (Wong et al., 2020). It has been estimated that up to 85% of the carbon footprint of vertical farms comes from their high electricity demands (Butturini & Marcelis, 2020), so it is paramount that vertical farms find the correct light recipes that effectively balance yield requirements and their carbon footprints.

For this reason, there is an immediate need to develop experimental designs that would prioritise the light qualities that are worth further investigation. We propose a three-phase workflow: (i) quantification of light heterogeneity within the facility (ii) an observational experimental design (iii) variable selection using lasso regularisation. The variables that are consistently selected and have high coefficients could be targeted for further investigation. Critically, we use cross-validation to ensure that our models are able to predict biomass in data that was not used to train the model, which ensures that our results are not simply a consequence of over-fitting. Moreover, this workflow can suggest areas where heterogeneity in microenvironments could be most impacting the yield of the crops, areas warranting technological improvements to ensure greater environmental refinement.

Our work highlights several key associations among light intensity, light quality and biomass. First, heterogeneity in light quality within a vertical farm may have a greater impact on uniform crop yields than heterogeneity in light intensity. This is consistent with (He et al., 2019), which finds that light quality influences leafy green properties, even under uniform light intensity conditions. This suggests that it is important for small-scale vertical farms to track the heterogeneity of light quality in their facilities, instead of solely

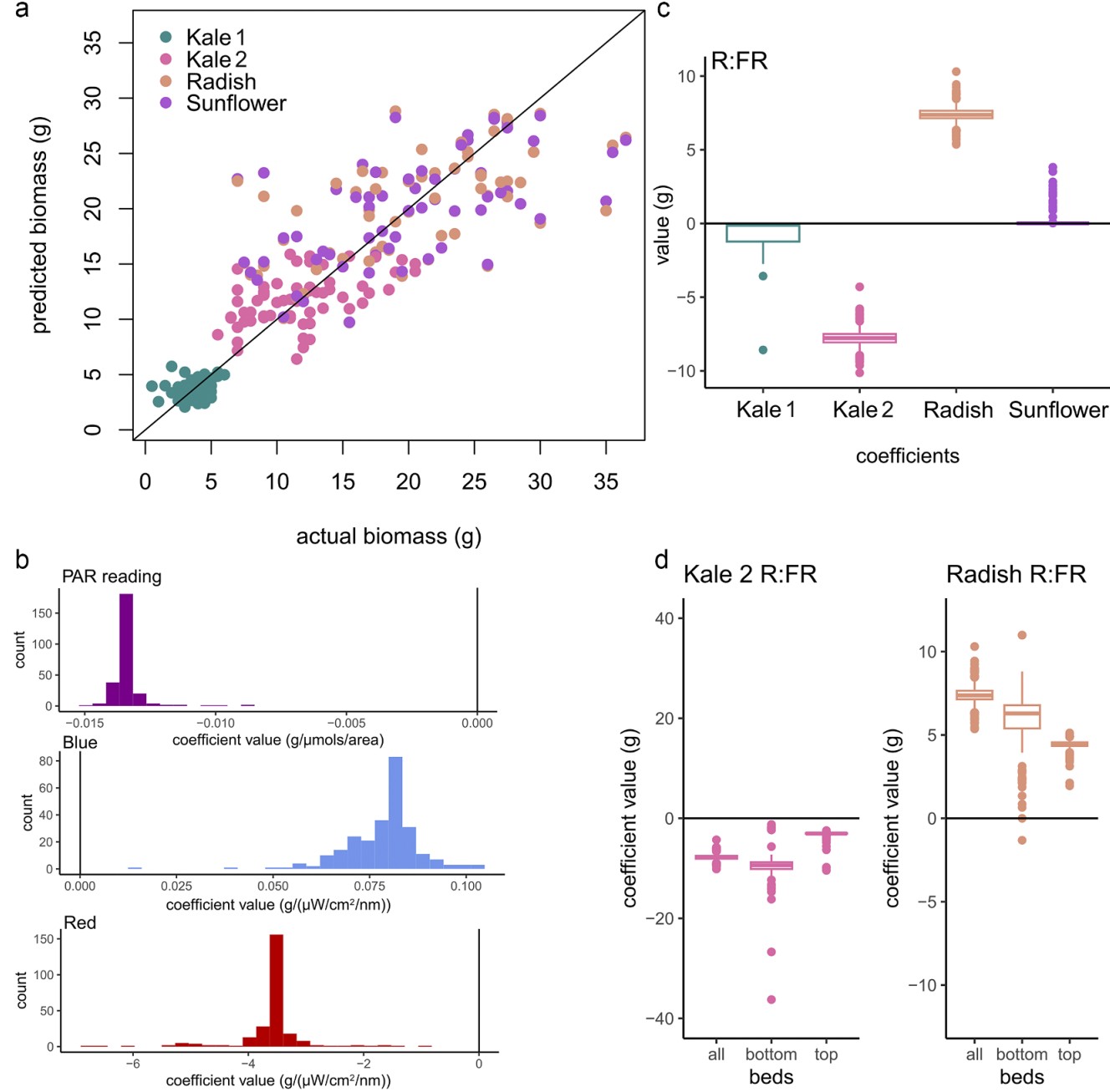

**Figure 3.** Analysis of lasso model coefficients. (a) Here, we compare the actual biomass at each position with the predicted biomass when the lasso model was trained using all the positions except that one (leave-one-out cross validation, LOOCV). Biomass always refers to the total biomass in each 83.7 cm² sampling site. (b) The histogram of coefficient values for the LOOCV lasso models, for three different parameters (PAR reading, blue and red). (c) The variety-specific coefficients for the R:FR ratio across the LOOCV lasso models. (d) The variety-specific coefficients for kale (Tozer) and radish, when different combinations of beds are used to train the model. Bottom refers to the bottom two beds, while top refers to the top two beds.

relying on PAR readings. Additionally, sensitivity to light ratios is likely to be variety-specific, which is consistent with several studies that have found different optimal light quality ratios were optimal for different varieties (Jin et al., 2023; Pennisi et al., 2019). This highlights how important it is to have ways of quickly screening light sensitivities in small-scale vertical farms, as a light recipe that works well for one variety is not guaranteed to produce optimal results for another. Other data sets in vertical farms have explored the role of light quantity and quality on yield, but these experiments were performed in a randomised controlled experimental design

and so fewer combinations of species and light combinations were tested, with many examples of vertical farm experiments only testing 3–5 different light recipes (Carotti et al., 2024; Dou et al., 2020; Semenova et al., 2022)

The specific light sensitivities of the microgreens that we highlight are likely to be specific to the vertical farm facility and microgreen varieties we have grown. Nevertheless, they indicate the kind of lessons that could be learnt from performing an observational study within a small-scale vertical farm. Our results have helped suggest light treatments that we can now test on our vertical farm

to improve yield and reduce energy consumption. An important caveat is that temperature and humidity heterogeneity were not quantified in our experiment. However, our experimental design could be extended to include these variables.

**Open peer review.** To view the open peer review materials for this article, please visit http://doi.org/10.1017/qpb.2025.10003.

**Supplementary material.** The supplementary material for this article can be found at http://doi.org/10.1017/qpb.2025.10003.

**Data availability statement.** All data and code to produce all figures and statistical tests in the manuscript are available at https://github.com/stressedplants/VerticalFarmOptimisation.

## Acknowledgements

We would like to acknowledge A. L. Tozer Limited for providing us with the kale seed. Additionally we would like to thank Paul Scott, Jason Daff, Harry Stevens, Alison Fenwick, Jacob Woodward, Dave Grimshaw and Peter Smithson of the University of York Department of Biology horticulture department for their assistance and support with this work. The research presented in this paper was conducted while PS was employed by Vertically Urban; PS is currently affiliated with The UK Agri-Tech Centre. The authors would like to thank Vertically Urban for their support during the research period. We also thank Spark:York CIC for their support of the Grow It York farm.

**Author contributions.** W.C. and D.E. conceived the study. W.C., D.E., P.S. and K.D. designed the study. W.C., A.K. and A.T. performed the experiments. W.C., E.R., G.V. and D.E. performed data analysis. W.C. and D.E. wrote and compiled the manuscript with contributions to the text from P.S., K.D., A.K., A.T., G.V. and E.R.

**Funding statement.** We would like to acknowledge the following funding sources: the Royal Society (RGS/R2/212345: D.E.), Biotechnology and Biological Sciences Research Council (Responsive Mode) (BB/V006665/1: D.E. and W.C.), the Biotechnology and Biological Sciences Research Council (White Rose Doctoral Training Partnership) (BB/T007222/1: E.J.R. and G.Y.W.V.), GenerationResearch and the Department of Biology (W.C., MRes studentship, https://generationresearch.ac.uk/) and FixOurFood - UKRI Strategic Priority Fund Transforming UK Food Systems project FixOurFood (BB/V004581/1: K.D.).

**Competing interests.** A.L. Tozer Limited supplied the seeds and Vertically Urban Limited provided the lighting.

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
