## [Reviewer Report]

The author may address the clarity of the data set taken into their research and how it is related to the other research work to compare the results.

The author may list out some pictures from the sample data.

How the cross-validation strategy is adapted to your research?

---

## [Reviewer Report]

In this paper the authors present a method by which heterogeneity within a small vertical farming setup can be used to train and optimize models in order to highlight which of a range of growth variables have the greatest impact within the specific system with the given plants.

The authors first present a baseline model, which assumes that physical location within the vertical farm is the only variable influencing plant growth. Three iterative models (equations 2-4), which introducing light intensity, light quality or light quality ratios as variables. These models were then used to compare back to the baseline in order to determine which of these factors has the greatest impact upon growth. Through this method, the authors determine that, independent of the other lighting factors, light quality plays a greater role in growth than light intensity or light quality ratios.

A fifth equation was then developed, combining all factors from the prior equations. This equation was used to train a population of 256 models each based on 255 of 256 observations of growth under the heterogeneous conditions in the vertical farm. Each of these models was evaluated by comparing the model to the remaining unfitted observation.

This population of models was then used to determine which variables were most consistently required to estimate biomass. This showed that light intensity, as well as blue and red light quality were consistently required for all models.

Light quality ratios were not independently able to predict biomass across all models but when assessing observations of some specific plant varieties R:FR ratio was shown to have significant but varied roles, making this an example of a key variable highlighted by the model for refinement if working with these specific crops.

Major comments

Please define the following terms:

Vertical farming

Taguchi method

Could a reference please be added following the first sentence of the materials and methods section: “This work was performed in the Grow It York vertical farm (located in York city centre, UK) which uses LettUs Grow (Bristol, UK) aeroponic technology.”

Please list the seed varieties used and where the seed used in the study was sourced from in the materials and methods section

Could the conversion from irradiance to resulting wattage be included in Table S1? I’m not convinced of the need to move between uE and uW within the models and it will likely be easier to use consistent units throughout.

In Materials and Methods section ‘Regularizing general linear modelling’ the authors state; “Data was standardised prior to fitting” please elaborate on how this standardisation was performed.

Should the results header “Variable prioritization suggests R:FR ratio is a target area in kale and rocket production” refer to radish, not rocket?

---

## [Editor Report]

Dear Authors

Sorry for keeping your waiting. One reviewer overdue for 2 weeks that caused the delay. 

As you can see from the reviews, both reviewers recommended minor revision. 

Please revise the manuscript accordingly. 

Thank you.

Boon Leong Lim

Editor

Professor

School of Biological Sciences

University of Hong Kong

Webpage: https://boon-leong-lim-lab.webflow.io/

---

## [Reviewer Report]

This paper is based on the concept that one of the principal disadvantage of vertical farm is light heterogeneity, and it can be turned in an advantage for crop cultivation. My major critical issues regarding this paper are:

- Regarding Abstract and Introduction, these sections do not sound scientifically speaking, as in the abstract only the objective of the research is presented and in the Introduction there is a complete lack of numerical data regarding previous literature.

- In the methods section, several points are not clear: why this specific combination of light has been chosen, and how the effectiveness of the light was assessed on the biomass growth. It is totally unclear to me how it is possible to establish any correlation regarding light and plants yield without evaluating the plants' growth.

- in the Discussion section, several paragraphs are more suitable for the introduction, as previous literature only is discussed.

- Conclusion section is missing but the final sentence of the abstract hints to results that are very well-known

---

## [Editor Report]

Congratulations, I have assessed your revised manuscript, and I am pleased to report that it is now acceptable for publication.